# Rosavin Ameliorates Hepatic Inflammation and Fibrosis in the NASH Rat Model via Targeting Hepatic Cell Death

**DOI:** 10.3390/ijms231710148

**Published:** 2022-09-05

**Authors:** Reda Albadawy, Amany Helmy Hasanin, Sara H. A. Agwa, Shaimaa Hamady, Yasmin M. Aboul-Ela, Mona Hussien Raafat, Samaa Samir Kamar, Mohamed Othman, Yahia A. Yahia, Marwa Matboli

**Affiliations:** 1Department of Gastroenterology, Hepatology & Infectious Disease, Faculty of Medicine, Benha University, Benha 13518, Egypt; 2Clinical Pharmacology Department, Faculty of Medicine, Ain Shams University, Cairo 11566, Egypt; 3Clinical Pathology and Molecular Genomics Unit, Medical Ain Shams Research Institute (MASRI), Faculty of Medicine, Ain Shams University, Cairo 11382, Egypt; 4Department of Biochemistry, Faculty of Science, Ain Shams University, Cairo 11566, Egypt; 5Histology and Cell Biology Department, Faculty of Medicine, Ain Shams University, Cairo 11566, Egypt; 6Histology and Cell Biology Department, Kasralainy Faculty of Medicine, Cairo University, Giza 12613, Egypt; 7Gastroenterology and Hepatology Section, Baylor College of Medicine, Houston, TX 77030, USA; 8Biochemistry Department, Faculty of Pharmacy, Misr University for Science and Technology, Giza 12566, Egypt or; 9Chemistry Department, School of Science and Engineering, American University in Cairo, New Cairo 11835, Egypt; 10Medical Biochemistry and Molecular Biology Department, Faculty of Medicine, Ain Shams University, Cairo 11566, Egypt

**Keywords:** rosavin, non-alcoholic steatohepatitis, hepatic cell death, hepatic, inflammation, cell detachment, oxidative stress, non-coding RNA

## Abstract

Background: Non-alcoholic fatty liver disease (NAFLD) represents the most common form of chronic liver disease that urgently needs effective therapy. Rosavin, a major constituent of the Rhodiola Rosea plant of the family Crassulaceae, is believed to exhibit multiple pharmacological effects on diverse diseases. However, its effect on non-alcoholic steatohepatitis (NASH), the progressive form of NAFLD, and the underlying mechanisms are not fully illustrated. Aim: Investigate the pharmacological activity and potential mechanism of rosavin treatment on NASH management via targeting hepatic cell death-related (*HSPD1/TNF/MMP14/ITGB1*) mRNAs and their upstream noncoding RNA regulators (*miRNA-6881-5P* and *lnc-SPARCL1-1:2*) in NASH rats. Results: High sucrose high fat (HSHF) diet-induced NASH rats were treated with different concentrations of rosavin (10, 20, and 30 mg/kg/day) for the last four weeks of dietary manipulation. The data revealed that rosavin had the ability to modulate the expression of the hepatic cell death-related RNA panel through the upregulation of both (*HSPD1/TNF/MMP14/ITGB1*) mRNAs and their epigenetic regulators (*miRNA-6881-5P* and *lnc-SPARCL1-1:2*). Moreover, rosavin ameliorated the deterioration in both liver functions and lipid profile, and thereby improved the hepatic inflammation, fibrosis, and apoptosis, as evidenced by the decreased protein levels of IL6, TNF-α, and caspase-3 in liver sections of treated animals compared to the untreated NASH rats. Conclusion: Rosavin has demonstrated a potential ability to attenuate disease progression and inhibit hepatic cell death in the NASH animal model. The produced effect was correlated with upregulation of the hepatic cell death-related (*HSPD1*, *TNF*, *MMP14*, and *ITGB1*) mRNAs—(*miRNA-6881-5P*—(*lnc-SPARCL1-1:2*) RNA panel.

## 1. Introduction

Non-alcoholic fatty liver disease (NAFLD) has become a worldwide health problem with increased prevalence, and serious complications [1]. It ranges in severity from simple steatosis to nonalcoholic steatohepatitis (NASH) which can develop into hepatic cirrhosis, and hepatocellular carcinoma [2]. The clinical importance of NASH and the current lack of effective medications have sparked a great interest to identify relevant pathophysiologic mechanisms that can be the target for the discovery of novel therapies [3].

The pathogenesis of NASH is complex and includes the perturbation of several sophisticated mechanisms, such as host genetics, environmental factors, intestinal dysbiosis, innate immune activation, hepatic inflammation, cell death, or fibrogenesis with progressive cell damage [4]. Under lipotoxic conditions, the hepatic cellular damage mediated by oxidative stress results in the release of reactive oxygen species (ROS), cytokines, and several pro-inflammatory mediators [5], which stimulate innate immune cells and induce the expression of liver vascular adhesion molecules [6]. Interactions between activated integrins and adhesion molecules on leukocytes [7] promote the loss of the cell barrier integrity and contribute to the cell detachment from the extracellular matrix [8], collectively resulting in apoptosis by a phenomenon called anoikis (a particular mechanism of cell death triggered by cell detachment from the extracellular matrix) [9]. If extensive cell death is not adequately corrected by the liver regenerative activity, acute liver damage may progress into liver failure and activation of the fibrogenic response [8]. Given the key role of cell death, apoptosis, in the pathology of NASH, it is a logical progression to target key players in anoikis cell death for potential medical therapies for this disease.

Increasing evidence has also highlighted the implicating role of epigenetic mechanisms, including noncoding RNAs (ncRNAs), in the pathogenesis of human NASH [10]. The main ncRNA classes, long ncRNAs (lncRNAs) and micro RNAs (miRNAs) translate alterations in metabolism and nutrients into a heritable pattern of gene expression [11]. Moreover, these ncRNAs are playing pivotal roles in inflammation, insulin signaling, and metabolism [12,13] and may facilitate the identification of non-invasive biomarkers or the development of therapeutic strategies for NASH [10]. Clearly, emerging data also demonstrates the critical role of bioinformatic analysis to identify candidate RNA species as biomarkers for NASH screening, diagnosis, and therapy [11].

Interestingly, a number of in vitro and in vivo studies have reported that the Rhodiola Rosea (R. Rosea), a member of the plant family Crassulaceae, possesses various protective effects such as anti-antioxidant, anti-inflammatory, and hepato-protective effects [14,15,16]. The recorded regulatory mechanisms, standing behind these effects, include inhibiting the expression of apoptotic genes, reducing cell death [17], and modulating miRNA expression [18]. This suggests that the non-coding RNA may be a target for Rhodiola Rosea L. treatment. One of the major-specific constituents of R. Rosea is Rosavin but few studies have focused on the biological activity of this compound [17].

Therefore, the present study aimed to investigate the pharmacological activity and potential mechanism of Rosavin treatment on NASH management via targeting hepatic cell death-related (*HSPD1/TNF/MMP14/ITGB1*) mRNAs and their upstream noncoding RNA regulators (*miRNA-6881-5P* and *lnc-SPARCL1-1:2*) in NASH animal model.

## 2. Results

### 2.1. Rosavin-Protein Interaction Analysis by Blind Docking

Rosavin recorded the highest affinity towards *TNF-α* with binding energy of −11.50 kcal/mole (Figure 1B) and the lowest affinity towards Integrin-beta 1 with binding energy of −7.50 kcal/mole. *HSPD1* and *MMP14* recorded affinities with binding energies of −8.30 and −7.90 kcal/mole, respectively. The top five poses are recorded for every protein in Appendix A. The specific interaction of rosavin with defined amino acids with bond lengths is shown in Figure 1 for all poses as Appendix A.

### 2.2. Rosavin-miRNA-6881-5P Analysis

The sequence of human *miRNA-6881-5P* was obtained from miRbase database (https://www.mirbase.org/, accessed on 22 July 2022) with accession MI0022728 [19]. The optimum secondary structure folded with a minimum free energy of −0.40 kcal/mole. The free energy of the thermodynamic ensemble was −0.71 kcal/mole and base pair dot plot probability is supplied as Appendix A. The tertiary structure prediction of *miRNA-6881-5P* predicted with RNA Composer. Using RLFold, the root-mean-square deviation (RMSD) between Rosavin poses and *miRNA-6881-5P* was selected to be 0.000. The lowest binding energy recorded was −39.406 kcal/mole for a pose of Rosavin miRNA-6881-5P interaction (Figure 1). The top five poses miRNA-Rosavin interaction are reported as Appendix A. Lennard-Jones potentials, electrostatic potentials, polarization, and H-bonding energies are reported in Appendix A.

### 2.3. The Effect of Rosavin Treatment on Liver Function and Lipid Profile

As illustrated in Table 1, compared to the normal control group (NC), there were significant increases (*p* < 0.001) in levels of serum AST, ALT, total bilirubin, direct bilirubin, and AFP with a significant decrease in the level of serum albumin upon feeding the experimental rats with HSHF diet in NASH group. Rats treated with rosavin in its three doses showed a significant reduction (*p* < 0.001) in these variables compared to animals in the untreated NASH group in a dose-dependent manner. Regarding lipid profile, a highly significant elevation in the serum levels of TG, TC, and LDL-C coupled with a significant decrease in the level of serum HDL-C in animals in the NASH group compared to the NC group. Daily administration of rosavin for four weeks, caused a significant decrease in the levels of serum TG, TC, and LDL-C associated with a highly significant increase in serum HDL-C, compared to the NASH group. Obviously, this ameliorative effect was more pronounced when rosavin was administered at the dose of 30 mg/kg.

### 2.4. Histological Findings

Grossly, the livers of rats fed the HSHF for 14 weeks showed a yellowish color, appeared enlarged, and became harder as compared to the normal control group. Liver tissue of HSHF-fed rats administered with rosavin at the dose of 10 mg/kg appeared redder and lighter than those of the NASH group.

Histological examination of the liver of the rats of the NASH group showed severe parenchymal damage, multi-lobular necrosis, dilated central veins, ballooned hepatocytes, obvious micro-vesicular and macro-vesicular steatosis, and substantial fibrosis. However, micro-vesicular steatosis accounted for most of the steatosis. Occasional ballooned hepatocytes displayed Mallory-Denk bodies were noted. The mean area % of collagen deposition displayed a significant increase as compared to the NC group. The RSV-10 group illustrated focal areas of inflammation, hepatocytes vacuolation, dilated central vein, and peri-portal fibrosis. The RSV-20 group displayed preserved architecture with some hepatocytes displaying variable cytoplasmic vacuolation with fine collagen deposition. Meanwhile, the RSV-30 group showed preserved architecture with most of the hepatocytes illustrating acidophilic cytoplasm and vesicular nuclei with prominent nuclei and fine collagen fibers (Figure 2 and Figure 3). All three rosavin-treated groups showed significant decreases in the mean area % of collagen deposition as compared to the NASH group (Figure 3f).

Regarding the NASH histological evaluation, as compared to the NC group, NASH and RSV-10 groups revealed significant increases in the mean NASH activity score, with a significant difference between both groups. Meanwhile, the mean NASH activity scores of the rats treated with RSV-20 and RSV-30 were significantly decreased as compared to NASH and RSV-10 (Figure 2k).

### 2.5. The Effect of Rosavin Treatment on the Expression Profile of Hepatic HSPD1/TNF/MMP14/ITGB1/miRNA-6881-5P/lnc-SPARCL1-1:2 Panel

The expression level of the RNA panel was assessed in the liver tissue of the experimental rats (Figure 4). Results showed that there was an upregulation in the expression of *HSPD1 mRNA*, *TNF mRNA*, *MMP14 mRNA*, *ITGB1 mRNA*, *miRNA-6881-5P*, and *lncRNA lnc-SPARCL1-1:2* in the NASH group (~4.5–11 folds) compared to the NC group (*p* < 0.001). Meanwhile, the daily administration of rosavin at its two doses 20 and 30 (RSV-20 and RSV-30 groups) had reduced the significant increases in hepatic RNA panel expression manifested in untreated NASH group animals. RSV-10 showed a much less significant reduction in the expression of mRNAs (*MMP14* and *ITGB1*), *miRNA-6881-5P*, and *lnc-SPARCL1-1:2* compared to the other treated groups.

### 2.6. The Effect of Rosavin Treatment on the Protein Expression of Hepatic Caspase-3, TNF-α and IL-6

Immunohistochemistry (IHC) for caspase-3, TNF-α, and IL-6 in the NASH group showed widespread strong positive immunostaining (IS) that was significantly increased as compared to the NC group. In RSV-10 and RSV-20, significant decreases in the IS mean area % of caspase-3, TNF-α, and IL-6 were estimated when compared to the NASH group. Meanwhile, a substantial decrease in the IS mean area % of caspase-3, TNF-α, and IL-6 was detected in RSV-30 as compared to NASH and other treated groups but was comparable to the normal control group. (Figure 5 and Table 2).

### 2.7. Correlation among Hepatic IL-6, TNF-α, Caspase-3 and Cell Death-Related RNA Panel

A significant positive correlation (*p* < 0.001) was recorded among relative expression (RQ) of cell *death-related*
*HSPD1/TNF/MMP14/ITGB1/miRNA-6881-5P/Inc-SPARCL1-1:2 RNA* panel. Furthermore, there was a positive correlation (*p* < 0.001) between RQ values of cell death-related RNA panel and both pro-inflammatory cytokines (TNF-α and IL-6) and the apoptotic marker (Caspase-3). These results were coupled with the presence of a positive correlation between the hepatic content of Caspase-3 and the hepatic content of pro-inflammatory cytokines TNF-α and IL-6 (Table 3).

## 3. Discussion

Despite the noticeable prevalence of nonalcoholic steatohepatitis (NASH) and its risk of progression to cirrhosis and hepatocellular carcinoma (HCC), approved pharmacological compounds for NASH are still lacking [20]. The molecular determinants of this pathogenic progression remain undefined but emerging data demonstrated that apoptotic cell death, including anoikis, is a major driving force for liver inflammation, leading to liver injury, and fibrosis in NASH pathogenesis [21]. Furthermore, previous studies have shown that Rhodiola Rosea (R. Rosea) constituents, specially rosavin, have anti-apoptotic activity in several metabolic diseases [18]. Therefore, in the present work, we evaluated the prospective pharmacological role of Rosavin treatment on NASH management and its modulatory effects on hepatic cell death-related players; (*HSPD1/TNF/MMP14/ITGB1*) mRNAs, (*miRNA-6881-5P*) and (*lnc-SPARCL1-1:2*), that were retrieved from bioinformatics studies in NASH animal model.

In NASH, hepatocyte lipotoxicity originates from the accumulation of lipid intermediates that cause hepatocyte injury and induce specific signaling cascades, resulting in hepatocyte apoptotic cell death [22]. Hepatic lipotoxicity results in mitochondrial dysfunction with subsequent production of reactive oxygen species (ROS). One of the mitochondrial quality control mechanisms is mitochondrial unfolded protein response (UPR^mt^) which generates mitochondrial chaperones, including heat shock protein Family D Member (*HSPD1*), and proteases for repairing unfolded protein stress [23]. *HSPD1* assists in proteolytic degradation of misfolded proteins and performs many physiological functions but can also be pathogenic in various conditions. Previous reports highlight the implication of *HSP60* in human cancer development [24]. It was also reported that upregulation of *HSPD1* suppressed the activity of mitochondrial complex IV, resulting in an increase in ROS concentration in cardiac muscle [25]. Additionally, the excess ROS production also stimulates nuclear factor-kappa B (NF-κB) with the resulting production of inflammatory cytokines such as IL-1, IL-6, and TNF-α [26]. These proinflammatory cytokines further stimulate Kupffer cells to produce proinflammatory cytokines that amplify and sustain the inflammatory response in the liver human NASH [27].

The results of the current study are consistent with the previously published data [28] where feeding Wistar rats a HSHF diet for 14 weeks successfully developed the spectrum of clinical changes, including elevated levels of serum liver function tests, dysregulated serum lipid profile, along with the hepatic structural changes observed in human NASH. There were significant upregulations in the expression of hepatic HSPD1 and TNF mRNAs in the NASH group. The results were also supported by the data obtained from the histological and immunohistochemistry analysis which revealed a significant widespread strong positive immunostaining for TNF-α and IL-6, coupled with large areas of inflammatory cells infiltration in the liver tissue, as compared to the NC group.

Accumulating evidence also illustrates that activated Kupffer cells can aggravate liver injury resulting in the appearance of newly synthesized extracellular matrix (ECM) proteins; collagens which progressively develop into the fibrotic liver. This results in matrix metalloproteinases (*MMPs*) activation which are the main enzymes involved in ECM degradation [9]. MMPs, such as MMP14, increase the expression of liver vascular adhesion molecules. Interactions between activated integrins (including integrin 1; *ITGB1*) expressed on immune cells and adhesion molecules mediate loss of the vascular endothelial cell barrier integrity and the cell detachment from the extracellular matrix. [29]. Accordingly, in the current study, we also reported increased expression of *MMP14* and *ITGB1* mRNAs in NASH together with a significant upsurge in the mean area % of collagen fibers, in comparison with the NC group. Parallel studies have also confirmed that *MMP14* [30] and *ITGB1* [31] are upregulated in hepatic fibrosis, which is gradually reduced with recovery suggesting that their blocking could be a potential anti-inflammatory therapeutic strategy in NASH. 

Obviously, apoptotic cell death depends on the activity of caspases (cysteine-aspartic proteases). In hepatocytes, caspase 8 is subjected to proteolytic autoactivation, resulting in direct or indirect (via mitochondria) activation of caspases-3, 6, and 7, which are implicated in the last step of cell death [22]. Previous studies revealed that increased inactive caspase-3 was strongly related to hepatocyte apoptosis and NASH progression in livers of NASH patients [32] and animal models [33]. The previously discussed data further elucidated our findings where a significant increase in the expression of the cytoplasmic and nuclear caspase-3 was detected in the NASH group compared to the normal control group.

Overall, dying hepatocytes are sufficient to release stress signaling molecules that can affect neighboring hepatic cells and trigger a variety of responses that can induce an exuberant response that can result in amplifying tissue inflammation and scarring [34]. Therefore, targeting cell death may become one of the therapeutic strategies for the treatment of NASH.

In the present study, all previously detected disturbances in the NASH group were significantly corrected by daily treatment of rats with rosavin for four weeks. Previous research indicated that Rhodiola Rosea (R. Rosea) components, including rosavin, attenuate inflammatory damage through decreased production of pro-inflammatory cytokines, inhibition of NF-kB, and activation of antioxidant signaling pathways. Moreover, interesting studies have revealed that R. Rosea extract can enhance cellular immunity by inhibiting the expression of apoptotic genes and decreasing tissue apoptosis [17].

The findings of the present study were in accordance with the previously published data. The results showed that rosavin treatment significantly improved the liver function tests and lipid panel, decreased the hepatic expression of IL-6 and TNF-α proteins, and diminished the percentage and stage of steatosis, inflammation, and fibrosis as compared with the untreated NASH group. Meanwhile, rats in the rosavin-treated groups recorded a significant decrease in the expression level of *HSPD1*, *TNF*, *MMP14*, and *ITGB1* mRNAs. A substantial decrease in the mean area % of apoptotic caspase-3 was also detected in RSV-treated groups as compared to NASH. Noticeably, rosavin at its two doses of 20 and 30 had a dominant improving effect over RSV-10. The detected results suggest that the use of rosavin could improve hepatic tissue injury via inhibiting cell death-related insults.

Emerging evidence illustrates that microRNAs (miRNAs) modulation could be one of the regulatory mechanisms behind the pharmacological activity of rosavin [18]. miRNAs and long non-coding RNAs (lncRNAs) are highly confirmed to play a crucial role in mediating the progression of diseases, including NASH. An altered hepatic miRNA profile has been shown in NASH in both human and animal models [35,36] and more evidence demonstrated the role of miRNAs in liver functions [37]. Concerning lncRNAs, their role in inflammation-related diseases, especially NASH, was mentioned in detail in a recent review by Shabgah et al., [38] who presented multiple examples of NAFLD-related lncRNAs and their potential mechanisms contributing to disease progression.

Using Bioinformatics, the miRWalk database predicted the upstream noncoding RNA regulators, *miRNA-6881-5P* and *Inc-SPARCL1-1:2* for the selected four mRNAs (HSPD1/TNF/MMP14/ITGB1). The results showed that there was a significant increase in the expression level of both hepatic *miRNA-6881-5P* and *hepatic Inc-SPARCL1-1:2* in NASH animals, compared to normal controls. On the other hand, daily administration of rosavin for four weeks, has significantly reduced their hepatic expression, compared to the NASH group. The crucial role of miRNAs in immunity strongly suggests their association with the regulation of hepatic inflammation and fibrogenesis [39]. The immune responses elicited by the injured liver in NASH are also controlled by miRNAs through the implication of several signaling pathways including transforming growth factor beta (*TGF-β1*)-signaling, cytokine-signaling, and toll-like receptors (*TLRs*) signaling [36,40]. Regarding lncRNAs, several lncRNAs are increased in conjunction with liver inflammation and fibrosis, and analyses of these RNAs showed multiple pathways, including those involved in *TGF-β* and *TNF* signaling and extracellular matrix deposition [10,41]. LncRNAs can act as co-factors that modify the activity of transcription factor which specifically binds an enhancer, thereby inducing expression of the adjacent protein-coding gene. Moreover, the functional DNA elements embedded in lncRNA loci can activate proximal enhancers of the neighboring gene [42,43]. Herein, the functional enrichment analysis of both *miRNA-6881-5P* and *lnc-SPARCL1-1:2* showed their implication in inflammatory and fibrogenic pathways. Accordingly, to all previously discussed data, this can illustrate the elevated expression of these ncRNAs species and their target genes (*HSPD1/TNF/MMP14/ITGB1*) in the NASH model group in the current study. To the best of our knowledge and based on the available literature, neither *miRNA-6881-5P* nor *lncRNA lnc-SPARCL1-1:2* has been correlated with liver disease or other metabolic disorders to determine their actual implicating mechanisms.

Taken together, the experimental model of the current study hypothesized that (Figure 6) treatment of the NASH animals with rosavin for four weeks significantly downregulated the expression of hepatic *lncRNA lnc-SPARCL1-1:2*, resulting in the downregulation of hepatic *miRNA-6881-5P* miRNA concomitant with a decrease in the levels of hepatic cell death-related players including (i) *HSPD1* (Oxidative stress response), (ii) *TNF-a* and *IL-6* (Inflammatory response), (iii) *MMP14* and *ITGB1* (Cell detachment and anoikis cell death), and (iv) caspase-3 (Apoptosis). This eventually diminishes the pathological disturbances induced in the NASH animal, improving both liver functions and lipid profile, thereby ameliorating inflammation and fibrosis, and finally halting the hepatic cell death.

This study provides new insight into the molecular mechanisms of NASH pathogenesis and introduces valuable clues regarding the pharmacological effects of rosavin as an effective therapy for NASH management. However, limitations exist, where performing further mechanistic studies by utilizing the molecular assay on a wider molecular RNA network should be highly considered.

## 4. Material and Methods

### 4.1. Chemicals and Drugs

Cholesterol, urethane, and cholic acid were obtained from Ralin B.V. (Lijinbaan, The Netherlands). Rosavin was purchased from Aktin Chemicals (Cat. No. APC-380, Chengdu, China).

### 4.2. Experimental Animals and Design

The experimental study was conducted on male Wistar rats weighing 150–170 g and obtained from Nile Pharmaceuticals Company, (Cairo, Egypt). Animals were housed under well-ventilated temperature-controlled conditions (20 ± 2 °C), 12 h light/dark cycle with free access to water and normal rat chow. All protocols for animal care and experiments were approved by the Faculty of Medicine Benha University Research Ethics Committee (Ethical Approval Number; MoHP0018122017, 1017) in accordance with the guidelines of the Declaration of Helsinki. The animal model of NASH was developed by feeding the rats a high sucrose and high fat (HSHF) diet containing 70% normal pellets, 20% of lard, 10% sucrose, 1% cholesterol, and 0.25% cholic acid [28].

After a one-week acclimatization period, the rats were randomly distributed into five groups (n = 8 for each group); the normal control group (NC) was fed a normal pellet diet (NPD), the NASH model group was fed a HSHF diet for 14 weeks, Rosavin-10 group, Rosavin-20 group, and Rosavin-30 group. In the last three groups, the rats were fed HSHF for 14 weeks and received 10 mg, 20 mg, and 30 mg rosavin/kg body weight [44], respectively. The drug administration was performed intraperitoneally (IP) daily for the last 4 weeks of the experimental protocol (Figure 7).

### 4.3. Blood Sampling and Liver Tissue Collection

At the end of the 14th week, rats were fasted for 12 h and then anesthetized with an intraperitoneal injection of a single dose of urethane (1.2 g/kg) [2] before sacrificing. Blood samples were drawn from the retro-orbital vein for separation of serum and stored at −20 °C for further biochemical analysis. Intact livers were removed quickly and dissected. Part of the liver was directly stored at −80 °C for RNAs and protein assays while the residual parts were instantly fixed in 10% neutral buffered formalin for histological and immunohistochemical examinations.

### 4.4. Serum Biochemical Assays (Liver Function and Lipid Profile Markers)

Serum Alpha-fetoprotein (AFP) was measured by using a rat sandwich AFP ELISA Kit purchased from MyBioSource (Cat. No. MBS452901, San Diego SDS, San Diego, CA, USA). Levels of serum aspartate transaminase (AST), alanine transaminase (ALT), total and direct bilirubin, albumin, and Lipid profiles [Triglycerides (TG), Total cholesterol (TC), HDL cholesterol (HDL-C), and LDL cholesterol (LDL-C)] were assessed in serum using commercial kits according to the manufacturer’s instructions and performed by AU680 multifunctional biochemistry analyzer (Beckman Coulter Inc., Brea, CA, USA).

### 4.5. Hepatic Histological Evaluation

The buffered formalin-fixed liver samples were dehydrated in alcohol, embedded in paraffin wax, and cut into thin sections (4-μm thick). Tissue sections were subjected to histological stains; hematoxylin and eosin (H&E) and Masson’s trichrome stains to evaluate the morphological changes and collagen deposition, respectively. The liver sections were assessed according to the SAF system [45] which assesses steatosis, activity, and fibrosis. The activity is scored as the sum of lobular inflammation and hepatocyte ballooning scoring [46]. The scoring was estimated as the followings: (i) steatosis: 0 (none); 1 (mild; involvement of 5–33% of parenchyma); 2 (moderate; 33–66% involvement); 3 (severe; >66% involvement); (ii) lobular inflammations: 0 (none); 1 (mild); 2 (moderate); 3 (severe); (iii) hepatocyte ballooning: 0 (none); 1 (few ballooned cells); 2 (many cell-prominent ballooning); and (iv) fibrosis stage: 0 (none); 1 (perisinusoidal/pericellular fibrosis); 2 (as 1 with periportal fibrosis); 3 (as 2 with focal/extensive bridging fibrosis); 4 (cirrhosis). The evaluation was estimated by double-blinded histologists (M.D.). The NASH activity score was calculated and graded as the followings; NASH (Score ≥ 5), borderline (3–4) or no NASH (<3) [47].

Leica Qwin 500 C image analyzer (Cambridge, UK) was used to assess the followings: (i) The area percent (%) of collagen deposition for the Masson’s trichrome stained sections. 

### 4.6. Bioinformatics Set Up

#### 4.6.1. Construction of the RNAs-Based Panel

Based on our interest, the RNAs species that are implicated in NASH pathogenesis and related to hepatic cell death were screened. Firstly, the differentially expressed genes (DEGs) in NASH were screened from the Gene Expression Omnibus (GEO) at the National Center for Biotechnology Information (NCBI) (www.ncbi.nlm.nih.gov/geo/, accessed on 22 July 2021) [48,49]. From the retrieved DEGs, Heat shock protein Family D Member 1 (*HSPD1*), tumor necrosis factor (*TNF*), Matrix metalloproteinase 14 (*MMP14*), and integrin β1 (*ITGB1*) were selected as they are crucial players contributing to hepatic cell death through their embroilment either in the inflammatory response, oxidative stress, apoptotic process, anoikis, or cell adhesion (Appendix A). Secondly, miRWalk 3.0 (http://mirwalk.umm.uni-heidelberg.de/, accessed on 22 July 2021) was used to retrieve miRNAs that interact with the four selected mRNAs. It was found that miRNA-6881-5P could interact with the 4 selected mRNAs. Lastly, the interaction between the retrieved miRNA and lncRNA was predicted by using miRWalk 2.0. The lncRNA SPARCL1-1:2 was identified to be interacting with the retrieved miRNA-6881-5P. Finally, (*HSPD1/TNF/MMP14/ITGB1*) mRNAs—(*miRNA-6881-5P*)—( *lnc-SPARCL1-1:2*) RNA panel was constructed. It is noteworthy to mention that the constructed RNA panel was previously validated in our internationally published research on human NASH [4]. 

#### 4.6.2. Rosavin-Protein Interaction Analysis by Blind Docking

We used blind docking algorithm to predict the interaction between rosavin and dysregulated corresponding proteins from the selected mRNAs. One monomer from the human mitochondrial HSPD1 single ring was used for docking since they were all identical. The protein structure was obtained from Protein Data Bank (PDB) with identification (PDB ID: 7AZP) [50]. Integrin-beta 1 was obtained from PDB as alpha5-beta1 complex (PDB ID: 3VI4) [51]. However, docking was carried out against the two subunits of integrin-beta 1 separately. Tumor Necrosis Factor alpha (TNF-α) and Hemopexin-like domain of matrix metalloproteinase (MMP14) were obtained (PDB ID: 6X82 and 3C7X), respectively [52,53]. Rosavin structure was obtained with ChemSpider ID: 7,999,634 [54]. We used the OpenBabel (version 3.1.1) an open-source command-line software created by Geoff Hutchison in Pittsburg, PA, USA. (Github: https://github.com/openbabel/openbabel/releases/tag/openbabel-3-1-1) [55] and MGL Tools (Version 1.5.7) for protein and ligand preparation. Blind docking was carried out using Blind Docking Server (https://bio-hpc.ucam.edu/achilles, accessed on 22 July 2021) [56] which uses Autodock Vina [57] for docking with a more extensive algorithm on every amino acid with a chiral carbon. The top 5 poses for rosavin docked in every protein were selected based on the highest total affinity to the protein. Protein-ligand pose interaction was determined using Protein-Ligand Interaction Profiler (PLIP) web tool (https://plip-tool.biotec.tu-dresden.de/, accessed on 22 July 2021) [58]. The interaction was visualized using PyMol (version 2.5.3). Rosavin structure was obtained with ChemSpider ID: 7,999,634 [13].

#### 4.6.3. Rosavin-miRNA-6881-5P Docking Interaction

We obtained the secondary structure of miRNA-6881-5P using RNAFold web server under Vienna RNA tools (http://rna.tbi.univie.ac.at/, accessed on 22 July 2021) [59]. The secondary structure was used to predict the tertiary structure using RNA Composer modeling server (https://rnacomposer.cs.put.poznan.pl/, accessed on 22 July 2021) [60,61]. We used RLDock command line software for RNA-ligand docking. The five lowest energy ligand poses were selected from the optimum clustered poses. The interaction between miRNA-6881-5P and Rosavin was profiled and visualized using Discovery Studio Visualizer (version 21.1.0.20298, accessed on 22 July 2021) [62].

### 4.7. Total RNA Extraction (mRNA, miRNA and lncRNA)

Total RNA was isolated from the 50 mg of frozen liver tissue samples using miRNeasy Mini Kit (Cat. No. 217004, Qiagen, Helman, Germany) as per instructions provided by the manufacturer. The concentration and purity of total RNA were measured using NanoDrop (Thermo Scientific, Waltham, MA, USA); the purity of the extracted RNAs (A260/A280) was 1.8–2. The total RNA extracted from the liver tissues was immediately reverse transcribed into single-stranded complementary DNA (cDNA) using RT2 First Strand Kit (Cat. No. 330404, Qiagen) and miScript II RT (Cat. No. 218161, Qiagen), as per instructions provided by the manufacturer. The reaction was conducted in Thermo Hybaid PCR express Thermal Cycler (Thermo Fisher Scientific, Waltham, MA, USA). 

### 4.8. Quantitative Polymerase Chain Reaction (qPCR) 

The expression levels of HSPD1, TNF, MMP14, and ITGB1 mRNAs and lncRNA- SPARCL1–1:2 in the liver samples were assessed using a Quantitect SYBR Green Master Mix Kit (Cat. No. 204143, Qiagen, Helman, Germany) and RT2 SYBR Green ROX qPCR Mastermix (Cat. No. 330500, Qiagen, Helman, Germany), respectively. miRNA-6881-5P expression in liver samples was measured by using miScript SYBR Green PCR Kit (Cat. No. 218073, Qiagen, Germany); according to the manufacturing protocol. The GAPDH, and *SNORD72* were used as the housekeeping genes to normalize the raw data for the chosen mRNA, miRNA and lncRNA, respectively. All primer reagents used in this study were obtained from Qiagen, Germany (details in Appendix A).

Real-time (RT)-qPCR has been in an Applied Biosystems 7500 FAST RT-PCR system (Applied Biosystems, Foster City, CA, USA) thermal cycler. The thermal program for the SYBR Green-based qPCR was as follows: denaturation at 95 °C for 10 min; 45 cycles of denaturation for 15 s at 95 °C; followed by annealing for 30 s at 55 °C; and lastly, extension for 30 s at 70 °C. All reactions were performed in duplicate. The threshold cycle (Ct) values over 36 were considered negative expressions. The relative quantification of RNA expression was carried out using the Livak method, where RQ = 2^−ΔΔ*C*t^ [63].

### 4.9. Immunostaining of Hepatic IL-6, TNF-α and Caspase-3

Immunohistochemistry for the paraffin sections was carried out using rabbit anti-caspase 3 (dilution 1:500; Cat. No. NB100–56113, Novus Biologicals, Littleton, CO, USA) as a marker for apoptosis [64], rabbit anti-TNF-α (dilution 1:200; Cat. No. NB600–587, Novus Biologicals, Littleton, CO, USA) and mouse anti-interleukin-6 (IL-6) (dilution 1:400; Cat. No. ab9324, Abcam, Cambridge, UK) as markers of inflammation [47]. Antigen retrieval was performed by heating in citrate buffer then quenching of endogenous peroxidase was processed using a peroxidase blocker. The sections were incubated with the primary antibodies (Ab) in a humidity chamber overnight followed by incubation with the secondary Ab for 45 min. Visualization of the reaction was performed using a 3,3′-Diaminobenzidine (DAB) detection kit with counterstain by Meyer hematoxylin. The area % of the (+ve) immunostaining for caspase3, TNF-α and IL-6 was measured using Leica Qwin 500 C image analyzer. The measures were obtained from 10 non-overlapping low power fields/section in each group.

### 4.10. Statistical Analysis

Data were presented as the mean ± SD. The distribution normality of the data was achieved using Shapiro–Wilk test. Statistical differences among experimental groups were determined by one-way analysis of variance (ANOVA) using GraphPad Prism software, version 8.0 (San Diego, CA, USA). Tukey’s test method was utilized to compare differences between groups. Correlation test was performed using the Pearson correlation coefficient. *p* < 0.05 was considered to be statistically significant.

## 5. Conclusions

Rosavin has demonstrated a potential ability to attenuate NASH progression, inhibit hepatic cell death, and diminish the metabolic and pathological disturbances observed in the applied NASH animal model. The produced effect was correlated with upregulation of the hepatic cell death-related (HSPD1, TNF, MMP14, and ITGB1) mRNAs—(miRNA-6881-5P)—(lnc-SPARCL1-1:2) RNA panel.

## Figures and Tables

**Figure 1 ijms-23-10148-f001:**
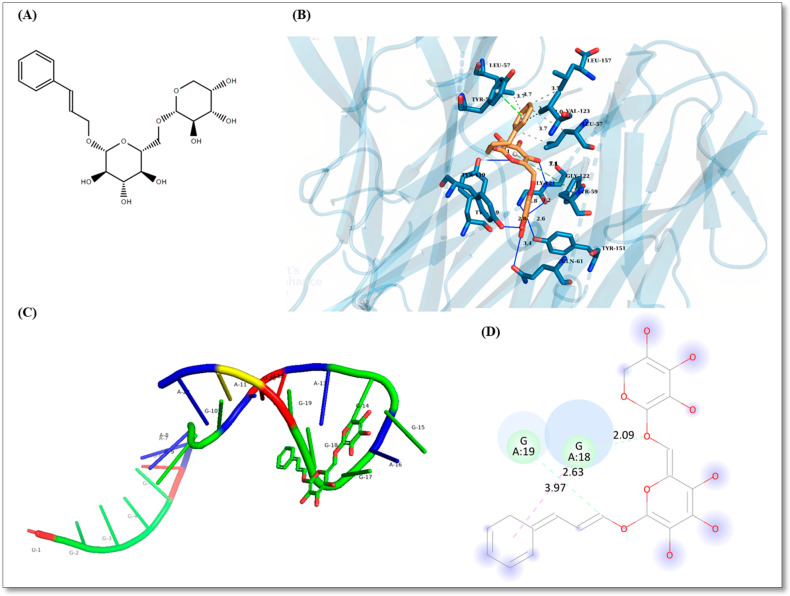
(**A**) Chemical structure of Rosavin. (**B**) Rosavin interaction with *TNF-α* with binding energy of −11.50 kcal/mole where green dashed-lines: pi-pi stacking, blue lines: hydrogen bonds, gray dotted-lines: perpendicular pi-pi stacking. Amino acids are expressed in three-letter nomenclature and interactions lengths are computed in Angstroms. (**C**) Rosavin highest affinity pose interacting with *miRNA-6881-5P*. (**D**) Rosavin interaction with *miRNA-6881-5P* where A indicates the sequence index, A for Adenosine, G for Guanosine, lemon dashed-line for carbon hydrogen interaction, red dashed-line for pi-pi stacking.

**Figure 2 ijms-23-10148-f002:**
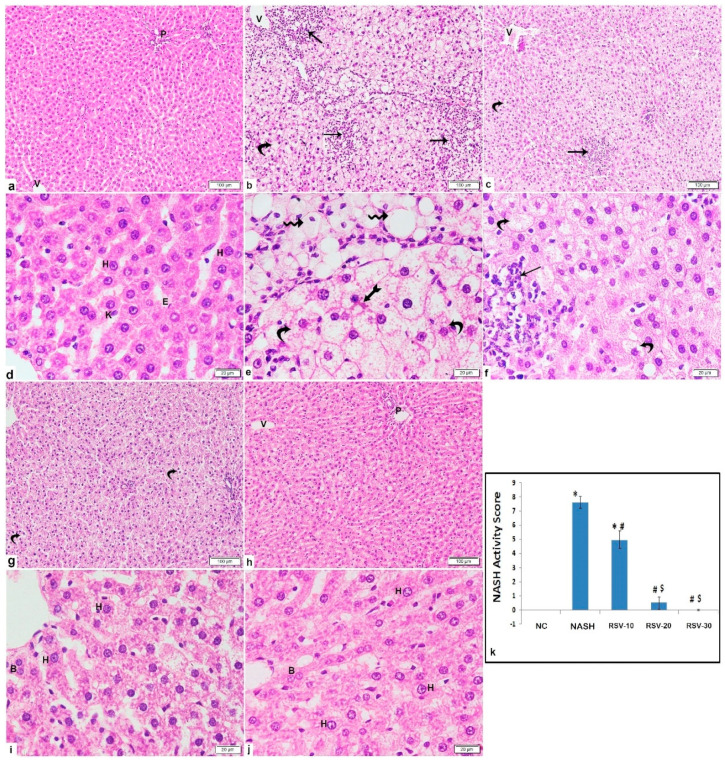
Photomicrographs of H&E-stained liver sections show: Photomicrographs of H&E-stained liver sections show: (**a**,**d**) NC: normal architecture displays portal tract (P), central vein (V) and radiating cords of hepatocytes separated by blood sinusoids. The hepatocytes illustrate acidophilic cytoplasm and vesicular nuclei with prominent nucleoli (H). The blood sinusoids are lined with endothelium (E) and Kupffer cells (K). (**b**,**e**) NASH: obvious damage of liver tissue with large areas of inflammatory cells infiltration (arrow), involving all zones; peri-central (zone3), mid-lobular (zone2) and peri-portal (zone1), widespread ballooning of hepatocytes illustrating micro-vesicular vacuolations (curved arrow) and dilated central veins (V). Some cells show one large-sized cytoplasmic vacuole (wavy arrow) with eccentric or peripheral flattened nuclei others display acidophilic cytoplasmic Mallory-Denk body (bifid arrow). (**c**,**f**) RSV-10: focal area of inflammatory cells infiltration (arrow) among the hepatocytes in zone2, micro-vesicular cytoplasmic vacuolation of hepatocytes (curved arrow) and dilated central vein (V). (**g**,**i**) RSV-20: preserved architecture with some hepatocytes displaying variable cytoplasmic vacuolation. The hepatocytes display acidophilic cytoplasm and either vesicular nuclei and prominent nucleoli (H), dark nucleus (arrowhead) or binucleation (B). (**h**,**j**) RSV-30: preserved architecture with central vein (V), portal tract (P) and most of hepatocytes illustrate acidophilic cytoplasm. (Inset): hepatocytes with vesicular nuclei and prominent nuclei (H) and binucleation (B). (**k**) The mean of NASH liver scoring (±SD) in the control and the experimental groups: * *p* < 0.05 compared to the NC; ^#^
*p* < 0.05 compared to NASH; ^$^
*p* < 0.05 compared to RSV-10. [Magnification: 100×; 400×].

**Figure 3 ijms-23-10148-f003:**
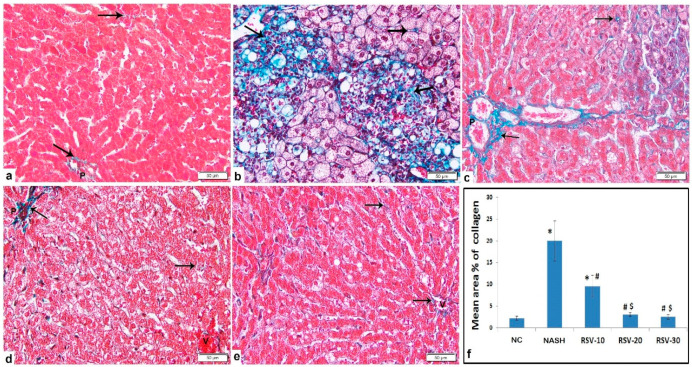
Photomicrographs of Masson trichrome-stained liver sections show: (**a**) Control: fine collagen fibers among hepatocytes and around the portal tract. (**b**) NASH: marked collagen deposition in multiple large areas in association with inflammatory cells infiltration and macro-vesicular steatosis, in addition to pericellular/perisinusoidal fibrosis. (**c**) RSV-10: increased peri-portal collagen deposition and among hepatocytes. (**d**,**e**) RSV-20 and RSV-30: fine collagen fibers. [Arrow: collagen deposition; P: portal tract; V: central vein]. [Magnification: 200×]. (**f**) The mean area % of collagen deposition (±SD) in the NC and the experimental groups: * *p* < 0.05 compared to the control group; ^#^ *p* < 0.05 compared to NASH; ^$^ *p* < 0.05 compared to RSV-10.

**Figure 4 ijms-23-10148-f004:**
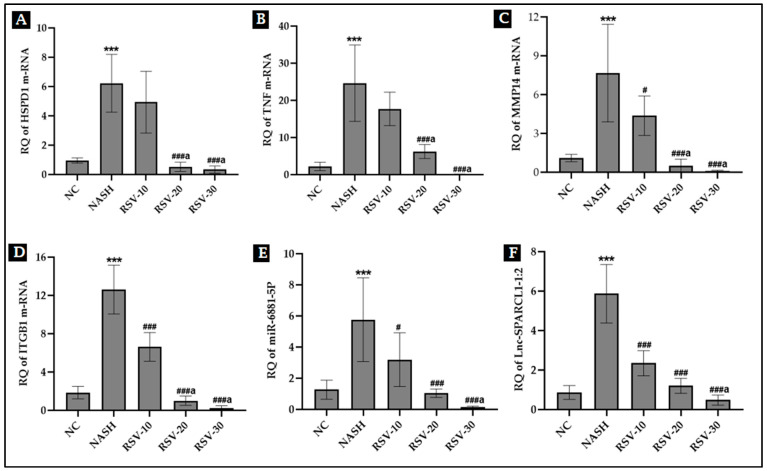
Effect of rosavin treatment on the expression level of hepatic m-RNAs: *HSPD1*, *TNF*, *MMP14*, and *ITGB1* (**A**–**D**); (**E**): hepatic *miRNA-6881-5P* and (**F**): hepatic *LncRNA SPARCL1*-1:2. Values are mean ± SD; number of animals = 8 rats/each group. **** p* < 0.001 vs. NC group; *^###^ p* < 0.001, and *^#^ p* < 0.05 vs. NASH group. *^a^ p* < 0.05 compared to RSV-10. One-way ANOVA followed by Tukey’s multiple comparison test RQ, relative quantification.

**Figure 5 ijms-23-10148-f005:**
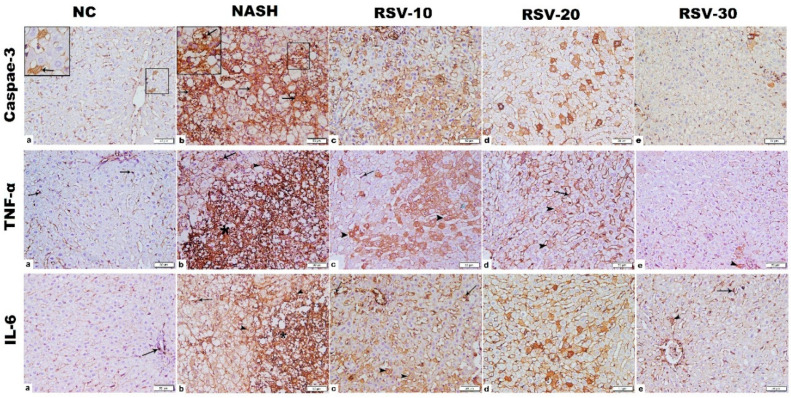
Photomicrographs of Caspase-3, TNF-α and IL-6 immunohistochemistry-stained liver sections. (1) Caspase-3; (**a**) NC group: few cells with cytoplasmic positive immunostaining (IS) were detected (inset), NASH group (**b**): widespread strong cytoplasmic, mainly, and nuclear (inset) positive IS. RSV-10 (**c**): many cells with cytoplasmic, mainly, and nuclear positive IS. RSV-20 (**d**): scattered cells with cytoplasmic positive IS. RSV-30 (**e**) showed few cells with weak cytoplasmic positive IS. (2) TNF-α; NC group (**a**): cytoplasmic positive immunostaining (IS) in macrophage lining sinusoid (arrow). NASH group (**b**): widespread strong cytoplasmic positive IS, mainly in the inflammatory cell infiltration (star), in addition to cells lining sinusoids (arrow) and hepatocytes (arrowhead). RSV-10 (**c**): cytoplasmic positive IS in numerous hepatocytes (arrowhead) in addition to cells lining sinusoids (arrow). RSV-20 (**d**): cytoplasmic positive IS in scattered hepatocytes (arrowhead) and cells lining sinusoids (arrow). RSV-30 (**e**): few hepatocytes with weak cytoplasmic positive IS (arrowhead). (3) IL-6; NC group (**a**): cytoplasmic positive immunostaining (IS) in few cells (arrow). NASH group (**b**): massive strong cytoplasmic positive IS, mainly in the inflammatory cell infiltration (star), in addition to cells lining sinusoids (arrow) and hepatocytes (arrowhead). RSV-10 (**c**): cytoplasmic positive IS in numerous hepatocytes (arrowhead) in addition to cells lining sinusoids (arrow). RSV-20 (**d**): cytoplasmic positive IS in scattered hepatocytes (arrowhead) and cells lining sinusoids (arrow). RSV-30 (**e**): few hepatocytes (arrowhead) and cells lining sinusoids (arrow) with cytoplasmic positive IS. [Magnification: 200×; inset: 400].

**Figure 6 ijms-23-10148-f006:**
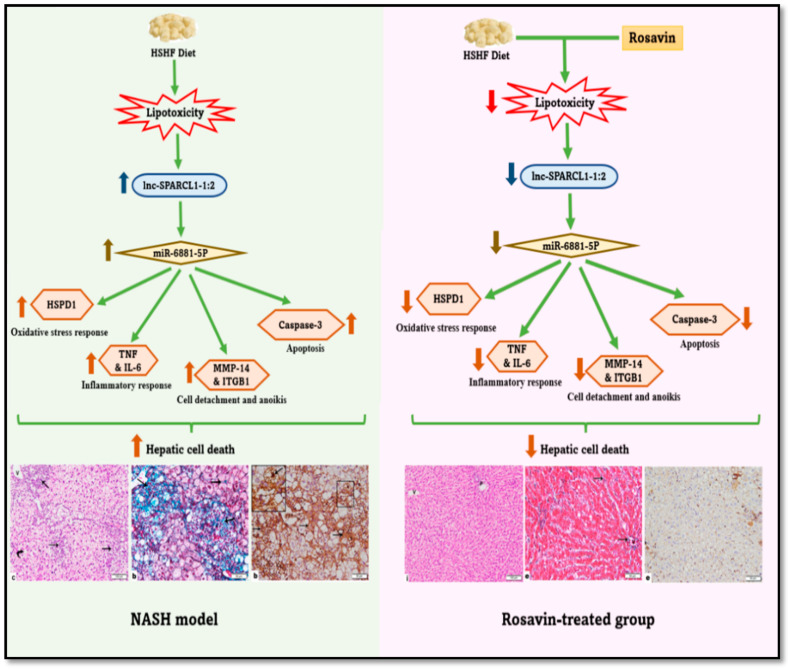
Schematic presentation of the study hypothesis.

**Figure 7 ijms-23-10148-f007:**
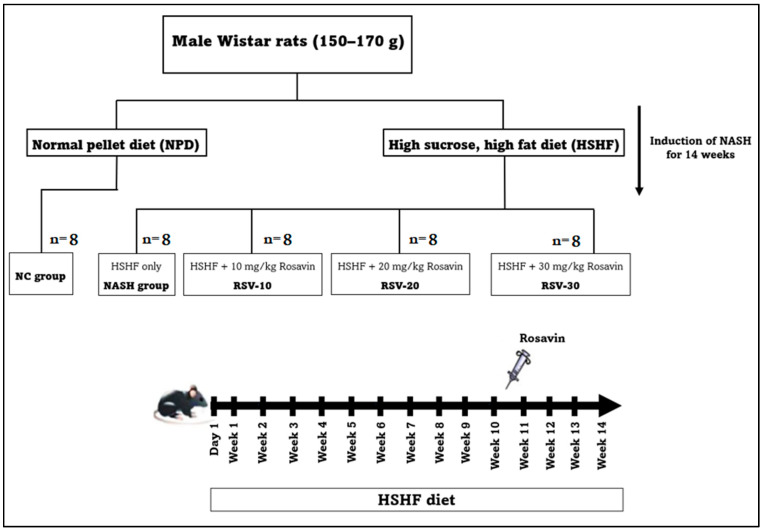
Schematic illustration of study design and the timeline for the animal experiment.

**Table 1 ijms-23-10148-t001:** The effect of rosavin treatment on liver function and lipid profile.

	NC	NASH	RSV-10	RSV-20	RSV-30
AST (IU/L)	21.17 ± 6.43	149.7 ± 36.49 *	88.17 ± 10.68 ^#^	57.67 ± 5.78 ^#a^	36.17 ± 5.04 ^#a^
ALT (IU/L)	10.1 ± 3.62	179.2 ± 38.86 *	83.05 ± 7.85 ^#^	45 ± 12.31 ^#a^	20.83 ± 6.11 ^#a^
T. bilirubin (mg/dL)	0.3967 ± 0.16	3.767 ± 0.47 *	2.65 ± 0.27 ^#^	1.635 ± 0.14 ^#a^	0.715 ± 0.19 ^#ab^
D. bilirubin (mg/dL)	0.2133 ± 0.08	2.567 ± 0.42 *	1.7 ± 0.33 ^#^	1.042 ± 0.15 ^#a^	0.495 ± 0.16 ^#ab^
Albumin (g/dL)	3.67 ± 0.37	1.68 ± 0.28 *	2.03 ± 0.25	2.60 ± 0.17 ^#a^	3.13 ± 0.49 ^#a^
AFP (ng/mL)	19.65 ± 3.43	540.2 ± 108.2 *	221.7 ± 32.16 ^#^	92.75 ± 20.5 ^#a^	29 ± 7.13 ^#a^
TG (mg/dL)	45.67 ± 4.89	173.7 ± 19.93 *	104 ± 22.09 ^#^	73.83 ± 11.44 ^#a^	55.17 ± 5.19 ^#a^
TC (mg/dL)	106.6 ± 13.92	228.1 ± 8.81 *	195.9 ± 7.90 ^#^	171.7 ± 6.93 ^#a^	118.2 ± 18.75 ^#ab^
HDL-C (mg/dL)	53.33 ± 3.72	21.67 ± 3.44 *	33.33 ± 1.97 ^#^	38.17 ± 1.17 ^#^	45.5 ±3.51 ^#ab^
LDL-C (mg/dL)	40.17 ± 10.5	171.7 ± 7.61 *	144.3 ± 8.48 ^#^	114.3 ± 12.14 ^#a^	72.17 ± 9.54 ^#ab^

Values are mean ± SD; number of animals = 8 rats/each group. * *p* < 0.001 compared to normal control group (NC). ^#^ *p* < 0.001 compared to NASH group, ^a^ *p* < 0.05 compared to Rosavin-10, ^b^ *p* < 0.05 compared to RSV-20. One-way ANOVA followed by Tukey’s multiple comparison test.

**Table 2 ijms-23-10148-t002:** The effect of rosavin on the mean area % of Caspase-3, TNF-α and IL-6 immunostaining.

	NC	NASH	RSV-10	RSV-20	RSV-30
Caspase-3	1.7 ± 0.3	45.3 ± 6.6 *	12.2 ± 3.8 *^#^	9.5 ± 2.4 *^#^	2.4 ± 0.8 ^# ab^
TNF-α	0.9 ± 0.2	39.7 ±7.1 *	11.4 ± 3.8 *^#^	9.3 ± 2.9 *^#^	1.4 ± 0.9 ^# ab^
IL-6	0.6 ± 0.1	44 ± 4.9 *	10.1 ± 4.6 *^#^	8.2 ± 3.4 *^#^	0.9 ± 0.3 ^# ab^

Values are expressed as means ±SD. * *p* < 0.05 vs. NC group; *^#^ p* < 0.05 compared to NASH group. ^a^
*p* < 0.05 compared to Rosavin-10, ^b^
*p* < 0.05 compared to RSV-20. One-way ANOVA followed by Tukey’s multiple comparison test.

**Table 3 ijms-23-10148-t003:** Pearson correlation among hepatic IL-6, TNF-α, Caspase-3 and cell death-related RNA panel.

	Caspase-3	TNF-α	IL-6	HSPD1	TNF	MMP14	ITGB1	miR-6881-5P	Lnc-SPARCL1-1:2
Caspase-3	1	0.95 *	0.97 *	0.74 *	0.8 *	0.77 *	0.89 *	0.79 *	0.93 *
TNF-α	0.95 *	1	0.97 *	0.75 *	0.77 *	0.77 *	0.9 *	0.77 *	0.9 *
IL-6	0.97 *	0.97 *	1	0.71 *	0.78 *	0.77 *	0.88 *	0.73 *	0.89 *
*HSPD1*	0.74 *	0.75 *	0.71 *	1	0.68 *	0.73 *	0.88 *	0.8 *	0.75 *
*MMP14*	0.77 *	0.77 *	0.77 *	0.73 *	0.78 *	1	0.77 *	0.85 *	0.85 *
*ITGB1*	0.89 *	0.9 *	0.88 *	0.88 *	0.79 *	0.77 *	1	0.81 *	0.89 *
*TNF*	0.8 *	0.77 *	0.78 *	0.68 *	1	0.78 *	0.79 *	0.65 *	0.8 *
*miR-6881–5P*	0.79 *	0.77 *	0.73 *	0.8 *	0.65 *	0.85 *	0.81 *	1	0.87 *
*Lnc-SPARCL1-1:2*	0.93 *	0.9 *	0.89 *	0.75 *	0.8 *	0.85 *	0.89 *	0.87 *	1

Number of observations = 30–50 * *p* < 0.001.

## Data Availability

All data generated during this study are included in this article.

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
