# Peer review of "Rosavin Ameliorates Hepatic Inflammation and Fibrosis in the NASH Rat Model via Targeting Hepatic Cell Death"

_ijms, 2022, doi:10.3390/ijms231710148_

Round 1

Reviewer 1 Report

Overall an interesting manuscript. Just a few points that need be addressed.

1. In introduction section, when talking about NASH pathogenesis, the following articles should be added as references: Byrne CD, Targher G. NAFLD: a multisystem disease. J Hepatol. 2015;62(1 Suppl):S47-S64. doi:10.1016/j.jhep.2014.12.012, regarding NAFLD pathogenesis, Lafoz E, Ruart M, Anton A, Oncins A, Hernández-Gea V. The Endothelium as a Driver of Liver Fibrosis and Regeneration. Cells. 2020;9(4):929. Published 2020 Apr 10. doi:10.3390/cells9040929 and Nasiri-Ansari, N.; Androutsakos, T.; Flessa, C.-M.; Kyrou, I.; Siasos, G.; Randeva, H.S.; Kassi, E.; Papavassiliou, A.G. Endothelial Cell Dysfunction and Nonalcoholic Fatty Liver Disease (NAFLD): A Concise Review. Cells 2022, 11, 2511. https://doi.org/ 10.3390/cells11162511, regarding endothelium damage in NAFLD

2. In abstract line 1 NASH should be replaced with NAFLD, since NAFLD is the most common form of liver disease worlwide 

3. In introduction section when talking about studies regarding rosavin, "limited studies" should be replaced with "few studies"

4. In results section, when describing histology, the inflammatory infiltrates in NASH mice should be characterized both in quantity and in location

5. Lastly, the discussion section is too large. It is unnecessary to describe again what anoikis is or talk about the lack of experiments regarding rosavin. Please make your discussion results-based.

Overall a noteworthy manuscript that should be accepted after minor, but necessary, revision

Author Response

Here is a point-by-point response to the reviewers’ comments and concerns.

Reviewer Comments to Author:

Reviewer #1:

Comments and Suggestions for Authors

  • In introduction section, when talking about NASH pathogenesis, the following articles should be added as references: Byrne CD, Targher G. NAFLD: a multisystem disease. J Hepatol. 2015;62(1 Suppl):S47-S64. doi:10.1016/j.jhep.2014.12.012, regarding NAFLD pathogenesis, Lafoz E, Ruart M, Anton A, Oncins A, Hernández-Gea V. The Endothelium as a Driver of Liver Fibrosis and Regeneration. Cells. 2020;9(4):929. Published 2020 Apr 10. doi:10.3390/cells9040929 and Nasiri-Ansari, N.; Androutsakos, T.; Flessa, C.-M.; Kyrou, I.; Siasos, G.; Randeva, H.S.; Kassi, E.; Papavassiliou, A.G. Endothelial Cell Dysfunction and Nonalcoholic Fatty Liver Disease (NAFLD): A Concise Review. Cells 2022, 11, 2511. https://doi.org/ 10.3390/cells11162511, regarding endothelium damage in NAFLD.

Author response: In response to the reviewer's inquiry, the recommended articles have been added as references in the introduction.

Introduction:

The pathogenesis of NASH is complicated and includes the disruption of several sophisticated mechanisms, such as host genetics, environmental factors, intestinal dysbiosis, innate immune activation, inflammation, cell death, or fibrogenesis with progressive liver damage [4]. Under lipotoxic conditions, the hepatic cellular damage mediated by oxidative stress results in the release of reactive oxygen species (ROS), cytokines, and other pro-inflammatory mediators [5], which activate innate immune cells and upregulate the expression of liver vascular adhesion molecules [6]. Interactions between activated integrins and adhesion molecules on leukocytes [7] promote the loss of the cell barrier integrity and contribute to the cell detachment from the extracellular matrix [8], collectively resulting in apoptosis by a phenomenon called anoikis (a particular mechanism of cell death triggered by cell detachment from the extracellular matrix) [9]. If extensive cell death is not adequately corrected by the liver regenerative activity, acute liver damage may progress into liver failure and activation of the fibrogenic response [8]. Given the key role of cell death, apoptosis, in the pathology of NASH, it is a logical progression to target key players in anoikis cell death for potential medical therapies for this disease.

References:

  1. Albadawy, R.; Agwa, S.H.A.; Khairy, E.; Saad, M.; El Touchy, N.; Othman, M.; Matboli, M. Clinical significance of hspd1/mmp14/itgb1/mir-6881-5p/lnc-sparcl1-1:2 rna panel in nafld/nash diagnosis: Egyptian pilot study. Biomedicines 2021, doi:10.3390/biomedicines9091248.
  2. Byrne, C.D.; Targher, G. NAFLD: A multisystem disease. J. Hepatol. 2015.
  3. Ibrahim, S.H. Sinusoidal endotheliopathy in nonalcoholic steatohepatitis: Therapeutic implications. Am. J. Physiol. - Gastrointest. Liver Physiol. 2021.
  4. Nasiri-ansari, N.; Androutsakos, T.; Flessa, C.; Kyrou, I.; Siasos, G.; Randeva, H.S.; Kassi, E.; Papavassiliou, A.G. Endothelial Cell Dysfunction and Nonalcoholic Fatty Liver Disease ( NAFLD ): A Concise Review. 2022.
  5. Lafoz, E.; Ruart, M.; Anton, A.; Oncins, A.; Hernández-Gea, V. The Endothelium as a Driver of Liver Fibrosis and Regeneration. Cells 2020.
  6. Duarte, S.; Baber, J.; Fujii, T.; Coito, A.J. Matrix metalloproteinases in liver injury, repair and fibrosis. Matrix Biol. 2015.

  • In abstract line 1 NASH should be replaced with NAFLD, since NAFLD is the most common form of liver disease worldwide.

Author response: : As suggested by the reviewer the NASH has been replaced with NAFLD in the abstract.

Abstract: Background: Non-alcoholic fatty liver disease (NAFLD) represents the most common form of chronic liver disease that urgently needs effective therapy. Rosavin, a major constituent of Rhodiola Rosea plant of family Crassulaceae, is believed to exhibit multiple pharmacological effects on diverse diseases. However, its effect on non-alcoholic steatohepatitis (NASH), the progressive form of NAFLD, and the underlying mechanisms are not fully illustrated. Aim: Investigate the pharmacological activity and potential mechanism of rosavin treatment on NASH management via targeting hepatic cell death-related (HSPD1/TNF/MMP14/ITGB1) mRNAs and their upstream noncoding RNA regulators (miRNA-6881-5P and lnc-SPARCL1-1:2) in NASH rats. Results: High sucrose high fat (HSHF) diet-induced NASH rats were treated with different concentrations of rosavin (10, 20, and 30 mg/kg/day) for the last 4 weeks of dietary manipulation. The data revealed that rosavin had the ability to modulate the expression of the hepatic cell death-related RNA panel through the upregulation of both (HSPD1/TNF/MMP14/ITGB1) mRNAs and their epigenetic regulators (miRNA-6881-5P and lnc-SPARCL1-1:2). Moreover, rosavin ameliorated the deterioration in both liver functions and lipid profile, and thereby improved the hepatic inflammation, fibrosis, and apoptosis, as evidenced by the decreased protein levels of IL6, TNF-α, and caspase-3 in liver sections of treated animals compared to the untreated NASH rats. Conclusion: Rosavin has demonstrated a potential ability to attenuate disease progression and inhibit hepatic cell death in the NASH animal model. The produced effect was correlated with upregulation of the hepatic cell death-related (HSPD1, TNF, MMP14, and ITGB1) mRNAs– (miRNA-6881-5P) – (lnc-SPARCL1-1:2) RNA panel.

  • In introduction section when talking about studies regarding rosavin, "limited studies" should be replaced with "few studies"

Author response: As suggested by the reviewer, "limited studies" has been replaced with "few studies" in introduction section. 

Introduction

Interestingly, a number of in vitro and in vivo studies have reported that the Rhodiola Rosea (R. Rosea), a member of the plant family Crassulaceae, possesses various protective effects such as anti-antioxidant, anti-inflammatory, and hepato-protective effects [14–16]. The reported regulatory mechanisms, standing behind these effects, include inhibiting the expression of apoptotic genes, reducing cell death [17], and modulating miRNA expression [18]. This suggests that the non-coding RNA may be a target for Rhodiola Rosea L. treatment. One of the major-specific constituents of R. Rosea is Rosavin but few studies have been focused on the biological activity of this compound [17]

  • In results section, when describing histology, the inflammatory infiltrates in NASH mice should be characterized both in quantity and in location.

Author response: regarding the quantification of the inflammatory infiltrates in NASH, it was assessed and quantified as a part of the SAF system of liver scoring estimated in our work as described by Bedossa Bedossa and Consortiumet, (2014). The scoring system assess steatosis, NAS activity and fibrosis. Activity is scored as the sum of lobular inflammation and hepatocyte ballooning scoring (Kleiner et al., 2005). The results already presented in the manuscript Figure 2k.

As recommended, the site of the inflammatory foci was clearly described and added to the legend of (Figure 2).

  • Bedossa, P.; Burt, A.A.; Gouw, A.H.A.; Lackner, C.; Schirmacher, P.; Terracciano, L.; Tiniakos, D.; Brain, J.; Bury, Y.; Cabibi, D.; et al. Utility and appropriateness of the fatty liver inhibition of progression (FLIP) algorithm and steatosis, activity, and fibrosis (SAF) score in the evaluation of biopsies of nonalcoholic fatty liver disease. Hepatology 2014, doi:10.1002/hep.27173.
  • Kleiner, D.E.; Brunt, E.M.; Van Natta, M.; Behling, C.; Contos, M.J.; Cummings, O.W.; Ferrell, L.D.; Liu, Y.C.; Torbenson, M.S.; Unalp-Arida, A.; et al. Design and validation of a histological scoring system for nonalcoholic fatty liver disease. Hepatology 2005, doi:10.1002/hep.20701

Figure 2. Photomicrographs of H&E-stained liver sections show: Photomicrographs of H&E-stained liver sections show: (a, d) NC: normal architecture displays portal tract (P), central vein (V) and radiating cords of hepatocytes separated by blood sinusoids. The hepatocytes illustrate acidophilic cytoplasm and vesicular nuclei with prominent nucleoli (H). The blood sinusoids are lined with endothelium (E) and Kupffer cells (K). (b, e) NASH: obvious damage of liver tissue with large areas of inflammatory cells infiltration (arrow) involving all zones; peri-central (zone3), mid-lobular (zone2) and peri-portal (zone1), widespread ballooning of hepatocytes illustrating micro-vesicular vacuolations (curved arrow) and dilated central veins (V). Some cells show one large-sized cytoplasmic vacuole (wavy arrow) with eccentric or peripheral flattened nuclei others display acidophilic cytoplasmic Mallory-Denk body (bifid arrow). (c, f) RSV-10: focal area of inflammatory cells infiltration (arrow) among the hepatocytes in zone2, micro-vesicular cytoplasmic vacuolation of hepatocytes (curved arrow) and dilated central vein (V). (g, i) RSV-20: preserved architecture with some hepatocytes displaying variable cytoplasmic vacuolation. The hepatocytes display acidophilic cytoplasm and either vesicular nuclei and prominent nucleoli (H), dark nucleus (arrowhead) or binucleation (B). (h, j) RSV-30: preserved architecture with central vein (V), portal tract (P) and most of hepatocytes illustrate acidophilic cytoplasm. (Inset): hepatocytes with vesicular nuclei and prominent nuclei (H) and binucleation (B). (K) The mean of NASH liver scoring (±SD) in the control and the experimental groups: *P<0.05 compared to the NC; # P<0.05 compared to NASH; $ P<0.05 compared to RSV-10. [Magnification: 100x; 400x].

  • Lastly, the discussion section is too large. It is unnecessary to describe again what anoikis is or talk about the lack of experiments regarding rosavin. Please make your discussion results-based.

Author response: As suggested by the reviewer, the discussion has been modified to be filtered from the repeated data and to be more concise.

Sincerely,

Dr Marwa Matboli, PhD

Reviewer 2 Report

Dear authors,

the manuscript at hand concerning the impact of rosavin treatment to a rat NASH model is well written and clear. The respective findings shed new light on the underlying mechanisms of rosavin-dependent effects and are therefore of interest. I have no major issues to be addressed, the only minor point is that the scale bars in the histologies should not be placed within the  pcitures but rather beside or under them.

Best Regards!

Author Response

Here is a point-by-point response to the reviewers’ comments and concerns.

Reviewer Comments to Author:

Reviewer #2:

Dear authors,

the manuscript at hand concerning the impact of rosavin treatment to a rat NASH model is well written and clear. The respective findings shed new light on the underlying mechanisms of rosavin-dependent effects and are therefore of interest. I have no major issues to be addressed, the only minor point is that the scale bars in the histologies should not be placed within the pictures but rather beside or under them.

Author response: We appreciate the reviewer’s feedback. It is noteworthy that the photomicrographing of the slides was performed using Olympus (Japan) digital camera connected to the light microscope and a computer system. The used software automatically adds a standard scale bar within the captured photo according to the selected magnification used.

Reviewer 3 Report

The authors demonstrate Rasavin protects against NASH in a rat model, by showing Rasavin reduction on hepatic oxidative stress response, inflammatory response, and anoikis/apoptosis. For mechanism study, authors firstly shown involvement of miRNA-6881-5p and lnc-SPARCL1-1:2 in the NASH. The reviewer think this part should be strengthened by providing results on 1) how miRNA-6881-5p and lnc-SPARCL1-1:2 are upregulated in the liver upon NASH condition; and 2)  how miRNA-6881-5p and lnc-SPARCL1-1:2 regulate downstream targets HSPD1/TNF/MMP14/ITGB1.

Author Response

Here is a point-by-point response to the reviewers’ comments and concerns.

Reviewer Comments to Author:

Reviewer #3:

The authors demonstrate Rosavin protects against NASH in a rat model, by showing Rasavin reduction on hepatic oxidative stress response, inflammatory response, and anoikis/apoptosis. For mechanism study, authors firstly shown involvement of miRNA-6881-5p and lnc-SPARCL1-1:2 in the NASH. The reviewer thinks this part should be strengthened by providing results on 1) how miRNA-6881-5p and lnc-SPARCL1-1:2 are upregulated in the liver upon NASH condition; and 2)  how miRNA-6881-5p and lnc-SPARCL1-1:2 regulate downstream targets HSPD1/TNF/MMP14/ITGB1

Author response: In response to the reviewer's inquiry, the manuscript has been modified to be included the mechanism of how miRNA-6881-5p and lnc-SPARCL1-1:2 are upregulated in the liver upon NASH condition; and 2) how miRNA-6881-5p and lnc-SPARCL1-1:2 regulate downstream targets HSPD1/TNF/MMP14/ITGB1. The data has been added to the discussion.

It is noteworthy that the present work is a preliminary step, and we are on our way to validating these results and plan to do in vitro functional analysis in further studies.

Discussion:

The crucial role of miRNAs in immunity strongly suggests their association with the regulation of hepatic inflammation and fibrogenesis [39]. The immune responses elicited by the injured liver in NASH are also controlled by miRNAs through the implication of several signaling pathways including transforming growth factor beta (TGF-β1)-signaling, cytokine-signaling, and toll-like receptors (TLRs) signaling [36,40]. Regarding lncRNAs, several lncRNAs are increased in conjunction with liver inflammation and fibrosis and analyses of these RNAs showed multiple pathways, including those involved in TGF-β and TNF signaling and extracellular matrix deposition [10,41]. LncRNAs can act as co-factors that modify the activity of transcription factor which specifically binds an enhancer, thereby inducing expression of the adjacent protein-coding gene. Moreover, the functional DNA elements embedded in lncRNA loci can activate proximal enhancers of the neighboring gene [42,43]. Herein, the functional enrichment analysis of both miRNA-6881-5p and lnc-SPARCL1-1:2 showed their implication in inflammatory and fibrogenic pathways. Accordingly to all previously discussed data, this can illustrate the elevated expression of these ncRNAs species and their target genes (HSPD1/TNF/MMP14/ITGB1) in the NASH model group in the current study.

Sincerely,

Dr Marwa Matboli, PhD

Round 2

Reviewer 3 Report

reviewer's questions are well addressed.